# Long-term outcomes of hospital survivors following an ICU stay: A multi-centre retrospective cohort study

Zakary Doherty[1,2]*, Rebecca Kippen[1], David Bevan[3], Graeme Duke[4], Sharon Williams[3], Andrew Wilson[5], David Pilcher[2,5,6,7]

1 School of Rural Health, Monash University, Melbourne, Victoria, Australia, 2 Alfred Health, Melbourne, Victoria, Australia, 3 Department of Health and Human Services, Melbourne, Victoria, Australia, 4 Eastern Health, Melbourne, Victoria, Australia, 5 Safer Care Victoria, Melbourne, Victoria, Australia, 6 The Australian and New Zealand Intensive Care Society (ANZICS) Centre for Outcome and Resource Evaluation, Melbourne, Victoria, Australia, 7 School of Public Health and Preventive Medicine, Monash University, Melbourne, Victoria, Australia

* zak.doherty@monash.edu

**Data Availability Statement:** Data cannot be shared publicly because it is the property of the Australian and New Zealand Intensive Care Society for which the study authors applied to for the use

## Abstract

### Background

The focus of much Intensive Care research has been on short-term survival, which has demonstrated clear improvements over time. Less work has investigated long-term survival, and its correlates. This study describes long-term survival and identifies factors associated with time to death, in patients who initially survived an Intensive Care admission in Victoria, Australia.

### Methods

We conducted a retrospective cohort study of adult patients discharged alive from hospital following admission to all Intensive Care Units (ICUs) in the state of Victoria, Australia between July 2007 and June 2018. Using the Victorian Death Registry, we determined survival of patients beyond hospital discharge. Comparisons between age matched cohorts of the general population were made. Cox regression was employed to investigate factors associated with long-term survival.

### Results

A total of 130,775 patients from 23 ICUs were included (median follow-up 3.6 years post-discharge). At 1-year post-discharge, survival was 90% compared to the age-matched cohort of 98%. All sub-groups had worse long-term survival than their age-matched general population cohort, apart from elderly patients admitted following cardiac surgery who had better or equal survival. Multiple demographic, socio-economic, diagnostic, acute and chronic illness factors were associated with long-term survival.

### Conclusions

Australian patients admitted to ICU who survive to discharge have worse long-term survival than the general population, except for the elderly admitted to ICU following cardiac surgery.

of the data. Data are available from ANZICS for those who meet the criteria for access to confidential data. Full ethics approval is required for any access to the data. To request access to data the ANZICS website (https://www.anzics.com.au/information-requests/) lists contact details and requirements.

**Funding:** The author(s) received no specific funding for this work.

**Competing interests:** The authors have declared that no competing interests exist.

These findings may assist during goal-of-care discussions with patients during an ICU admission.

## Background

Outcomes of patients admitted to Intensive Care Units (ICUs) in Australia have progressively improved over the past 18 years with reductions in hospital mortality for almost all patient groups [1, 2]. However, beyond discharge much less is known about the survival of ICU patients.

Previous investigations of the long-term survival of ICU patients have consistently shown that, compared to an age-matched cohort, ICU patients have shorter long-term survival, world-wide and in Australia [3–8]. However, the findings of these studies have limited generalisability due to small sample sizes, single-centre design, focus on specific patient groups or lack of important variables pertaining to the Intensive Care admission.

Recent work has highlighted the need to examine how factors before, during and after hospitalisation influence a patient's long-term trajectory [9]. This information can identify areas for future research, provide a benchmark for comparisons between different service providers and inform patient discussions regarding goals of care.

The aim of this study was to describe the epidemiology and quantify long-term survival of patients discharged alive following care in ICU in Victoria, Australia and to identify factors measurable while in hospital which were associated with time to death.

## Methods

### Study design

This retrospective cohort study linked three individual datasets: the Australian and New Zealand Intensive Care Society (ANZICS) Adult Patient Database (APD), the Victorian Admitted Episodes Dataset (VAED), and the Victorian Death Registry. The project was approved by the Alfred Health Human Research Ethics Committee (ref: 77/20), and the governance committee of the ANZICS Centre for Outcome and Resource Evaluation, and the Centre for Victorian Data Linkage.

### Data sources

**Australian and New Zealand Intensive Care Society Adult Patient Database.** The ANZICS APD is a bi-national clinical registry that contains de-identified information on all admissions to participating adult ICUs in Australia and New Zealand. Data are collected by trained ICU staff. All public sector adult ICUs in the State of Victoria, Australia contributed throughout the study period. Data captured include patient demographic and clinical characteristics (diagnostic, biochemical and physiological) from the first 24 hours of ICU admission required for the calculation of severity of illness scores (Acute Physiology and Chronic Health Evaluation (APACHE) III-J [10]) and information on patient outcomes.

**Victorian Admitted Episodes Dataset.** The VAED is an administrative dataset containing coded diagnostic (ICD-10AM) information, demographic data, and outcomes for all hospitalisations in Victoria, Australia. It is submitted to the Victorian Agency for Health Information by all Victorian hospitals [11]. Socioeconomic status and geographic remoteness of each patient's area of residence were included by matching listed area of residence in the

VAED with relevant data from the Australian Bureau of Statistics. The Index of Relative Socio-economic Advantage and Disadvantage (IRSAD), and the Australian Statistical Geography Standard Remoteness Structure were used to define socioeconomic status (SES) and geographic remoteness respectively [12, 13]. An Elixhauser Comorbidity score was derived for each patient based on the ICD-10AM codes assigned to them [14].

**Victorian Death Registry.** The Victorian Death Registry is an administrative dataset that records the date and cause of all deaths that occur in the study state [15]. Any death certificate issued in the state is logged in this dataset. Any deaths that occur outside of study state are not included in the registry.

## Study setting

The study was conducted in the State of Victoria, Australia, with a population of 5.2–6.5 million over the study period [16]. Victoria has a combination of public and private hospitals, with a total of 23 ICUs within 23 public hospitals. All were included in this study. Admissions to public ICUs represented 71% of the total ICU admissions in Victoria during the study period [17].

## Study population

All admissions for adults (≥18 years of age) to a public Victorian ICU between 1 July 2007 and 30 June 2018 (inclusive) were extracted from the APD. These admissions were linked with administrative data (VAED). Exclusions included readmissions for the same individual during the study period, admissions for palliative care or a potential organ donation, paediatric admissions (<18 years of age), patients with a usual residence outside the State of Victoria and patients who did not survive to hospital discharge. Patients aged 100 years or older at any time during their follow-up period were excluded from survival curves as single-years values are not available in Australian life tables from age 100. No sample size calculation was performed as this study was purely descriptive.

## Linkage process

Data linkage was undertaken by the Centre for Victorian Data Linkage [18]. The VAED and Victorian Death Registry were linked for deaths occurring up to 3rd August 2018, using patient identifiers such as name and date of birth. A combination of probabilistic and deterministic methods were employed to link de-identified patient records from the ANZICS APD to the VAED and to the Victorian Death Registry using age, gender, dates of hospital and ICU admission and discharge and hospital site. A de-identified dataset was then provided to the researchers. The match rate was calculated by dividing the admissions successfully matched after all inclusion criteria were applied (except readmissions) by all admissions after the same inclusion criteria were applied (*S1 Fig*). The readmission criteria were not applied as the variable for this was only available to matched admissions.

## Variables

Patient demographic variables include age, sex, relationship status, socioeconomic status (divided into IRSAD quartiles with the first quartile representing those with the lowest socioeconomic status), area of residence, comorbidities (quantified by an Elixhauser Comorbidity Score using the AHRQ weighting) and presence of a limitation of medical therapy order (defined as an order existing prior to ICU admission constraining medical treatment due to patient wishes or medical futility). Other variables included year of hospital admission

(combined into 3-year groups), hospital location, ICU admission type (elective or emergency), whether the admission was the result of a Medical Emergency Team call, source of admission to ICU, length of stay (LOS) in ICU and hospital, whether the patient was invasively ventilated or received renal replacement therapy, APACHE III-J score, discharge location and the ICU admission diagnosis group.

### Statistical analysis

Categorical variables were reported as frequencies with percentages and continuous variables as medians with interquartile ranges (IQR). Group differences were assessed using chi-square tests and the Mann-Whitney test as appropriate. All missing data were reported as such, no imputation was performed.

Age-stratified Kaplan-Meier survival curves with 95% confidence intervals were generated. The discharge date from hospital was used as time zero to prevent immortal time bias influencing results. Survival time was the number of days between time zero and the date of death (if present) or census date. Patients still alive on the 3rd August 2018 were right-censored. All figures include an additional curve demonstrating expected survival for the general population matched on year, age and sex to the study cohort represented in the curve. Australian life tables were used to produce these curves [19].

Cox proportional hazards regression was used to investigate factors associated with long-term survival. To account for independent effects of the variables on patient survival, patient age and the acute physiology sub-score of the APACHE III-J score were entered separately into the regression. The proportional hazards assumption was checked visually. The linearity assumption was checked by plotting continuous variables against their Martingale residuals. Patient age, APACHE III-J acute physiology score and hospital LOS were found to violate this assumption and were transformed into categorical variables. All variables were significant in univariable analysis and were thus included in the multivariable analysis. The largest group for each variable was chosen as the reference group except in the case of year of admission where the earliest year group was chosen. Hazard ratios and 95% confidence intervals (CI) were reported for each variable level relative to the reference group. Patients with any missing data were excluded from the Cox regression, except for the limitations of medical treatment variable where missing data were included as a separate level. This was done as in the early years of the study the completion of this variable was optional. All analyses were conducted using the *R Project for Statistical Computing Software Version 1.3.959.*

## Results

### Study cohort

After applying exclusion criteria (*S1 Fig)* the final study cohort consisted of 130,775 patients from 23 ICUs, Using the previously described method, 80.3% of eligible ICU admissions in the ANZICS APD were able to be matched to a VAED record. Comparisons between the study cohort and those that could not be linked are provided in *S1 Table*.

### Cohort characteristics

Characteristics of the study cohort are shown in *Table 1*. Patients had a median age of 64 years and over half were male (59%). They mainly lived in a major city (57%) or an inner regional town (30%). The study cohort were primarily from areas of slightly lower than average SES. Only 4% of patients had a limitation of medical therapy order in place on admission to ICU. The most common source of admission to ICU was from the Operating Theatre (48%)

**Table 1. Characteristics of the cohort (n = 130775).**

| Characteristic | N |
|---|---|
| **Age (years) (median, IQR)** | 64.0 (49.0, 74.8) |
| **Sex** | |
| Male | 77089 (58.9%) |
| Female | 53686 (41.0%) |
| **Relationship status** | |
| Partnered | 72193 (55.2%) |
| Not partnered | 54184 (41.4%) |
| Unknown | 4398 (3.3%) |
| **IRSAD score (median, IQR)** | 983 (937, 1037) |
| Unknown | 903 (0.6%) |
| **Patient area of residence** | |
| Major City | 74083 (56.6%) |
| Inner Regional | 39790 (30.4%) |
| Outer Regional | 11246 (8.5%) |
| Remote | 139 (0.1%) |
| Unknown | 5517 (4.2%) |
| **Elixhauser score (median, IQR)** | 2 (0,10) |
| **Admission post Medical Emergency Team call** | 12691 (9.7%) |
| Unknown | 5349 (4.0%) |
| **Limitations of medical therapy at ICU admission** | |
| No limitations | 121279 (92.7%) |
| Limitations | 4697 (3.5%) |
| Unknown | 4799 (3.6%) |
| **Year of admission** | |
| 2007–2009 | 25125 (19.2%) |
| 2010–2012 | 35793 (27.3%) |
| 2013–2015 | 38835 (29.6%) |
| 2016–2018 | 31022 (23.7%) |
| **Hospital type** | |
| Tertiary | 68878 (52.6%) |
| Metropolitan | 37101 (28.3%) |
| Rural / Regional | 24796 (18.9%) |
| **Admission type** | |
| Emergency | 84566 (64.6%) |
| Elective | 45332 (34.6%) |
| Unknown | 877 (0.6%) |
| **Admitted to ICU from** | |
| Theatre | 62785 (48.0%) |
| ED | 41086 (31.4%) |
| Ward | 16776 (12.8%) |
| Hospital Transfer | 10128 (7.7%) |
| **Length of ICU stay (days)** | 1.8 (0.9, 3.5) |
| Unknown | 20 (<0.1%) |
| **Length of hospital stay (days)** | 8.3 (4.7, 15.1) |
| Unknown | 2 (<0.1%) |
| **Invasively ventilated during admission** | 55303 (42.2%) |
| Unknown | 3945 (3.0%) |

(*Continued*)

**Table 1.** (Continued)

| Characteristic | N |
|---|---|
| **Invasively ventilated post elective surgery**[*] | 21454 (52.3%) |
| Unknown | 722 (1.7%) |
| **Duration of ventilation if applicable (days)** | 0.79 (0.4, 2.5) |
| Unknown | 3248 (5.9%) |
| **CRRT during admission** | 3906 (2.9%) |
| **APACHE III-J score (median, IQR)** | 49 (36, 64) |
| Unknown | 624 (0.4%) |
| **SOFA score at admission (median, IQR)** | 3 (2, 5) |
| **Discharged home** | 91700 (70.1%) |
| **Diagnosis on admission to ICU** | |
| Cardiac (Surgical) | 19997 (15.2%) |
| Cardiac (non-Surgical) | 8980 (6.8%) |
| Cardiac arrest | 2419 (1.8%) |
| Vascular/Thoracic surgery | 7966 (6.0%) |
| Abdominal aortic aneurysm | 2324 (1.7%) |
| Respiratory | 6477 (4.9%) |
| COPD | 2669 (2.0%) |
| Pneumonia | 5703 (4.3%) |
| Sepsis (excluding Pneumonia) | 9438 (7.2%) |
| Neurological | 3354 (2.5%) |
| Stroke / Intra-cerebral Haemorrhage | 1673 (1.2%) |
| Sub-arachnoid Haemorrhage | 1090 (0.8%) |
| Seizure | 1998 (1.5%) |
| Orthopaedic (non-spinal) | 4084 (3.1%) |
| Orthopaedic (spinal) | 631 (0.4%) |
| Trauma (head) | 3076 (2.3%) |
| Trauma (non-head) | 5953 (4.5%) |
| Toxicological (overdose) | 6275 (4.7%) |
| GI Surgery | 16976 (12.9%) |
| GI Medical | 4331 (3.3%) |
| Surgical (other) | 7976 (6.0%) |
| Medical (other) | 7385 (5.6%) |

Where unknown is not listed under a variable no data were missing.

[*]Calculation of proportion utilised elective surgery total as denominator.

IQR; Interquartile range, IRSAD; Index of Relative Socioeconomic Advantage and Disadvantage, ICU; Intensive Care Unit, ED; Emergency Department, LOS; Length of stay, CRRT; Continuous Renal Replacement Therapy, APACHE; Acute Physiological and Chronic Health Evaluation, SOFA; Sequential Organ Failure Assessment, COPD; Chronic Obstructive Pulmonary Disease, GI; Gastrointestinal.

followed by the Emergency Department (31%). During their admission just under half of patients were invasively ventilated with 3% requiring renal replacement therapy. The median APACHE III-J score was 49 (IQR: 36–64). The most common diagnosis group was cardiac (surgical) (15%) with gastrointestinal surgery following (13%). Patients spent a median of 1.8 days (IQR: 0.9–3.5) and 8.3 days (IQR: 4.7–15.2) in ICU and hospital, respectively. Seventy percent of patients were discharged home from hospital.

## Long-term survival

Follow-up ranged from 0 days (the patient died immediately following discharge) to 11.1 years post-discharge (for those admitted in 2007), with a median of 3.6 years post-discharge. A total of 25,046 (19%) patients were admitted to any public ICU during a subsequent hospital admission.

Survival at 1-year post-discharge was 90.2% (95% CI: 90.0% - 90.3%), lower than the expected survival at the equivalent time of 97.9% for the general population. Survival dropped at 5 years to 73.1% (95% CI: 72.8% - 73.4%) and at 10 years to 57.5% (95% CI: 57.1% - 58.0%). At the equivalent time points, expected survival for the general population was 89.5% and 79.1%. *Fig 1* shows the survival curve for the full cohort, *Fig 2* shows survival stratified by age group. For all age groups, survival was worse than the general population.

All diagnoses, with one exception, had worse long-term survival than the general age-matched population. Those admitted for cardiac surgery and aged over 70 years, had better survival than the general population up until 9 years post-discharge, after which survival was equal. For this group survival at 1-year post-discharge was 96.8% (95% CI: 96.4% - 97.2%), whereas for the general population it was 95.3%. At 5-years post-discharge, survival decreased to 81.5% (95% CI: 80.6% - 82.5%) and 76.5% for the cardiac surgery and general population, respectively. Chronic obstructive pulmonary disease (COPD) admissions had the worst long-term survival of all groups, in particular for the 55–70 years age group. At 5-years post discharge only 48.4% (95% CI: 45.1% - 51.9%) of the COPD group were alive compared to 78.8% of general age-matched population. The survival curves for select diagnosis groups are shown in *Fig 3*. Additional survival curves examining readmission status and admission type (elective surgery, emergency surgery or medical) are also included in the appendix (*S3 Fig*). All survival curves can be accessed in an interactive format at *iculongterm.com*

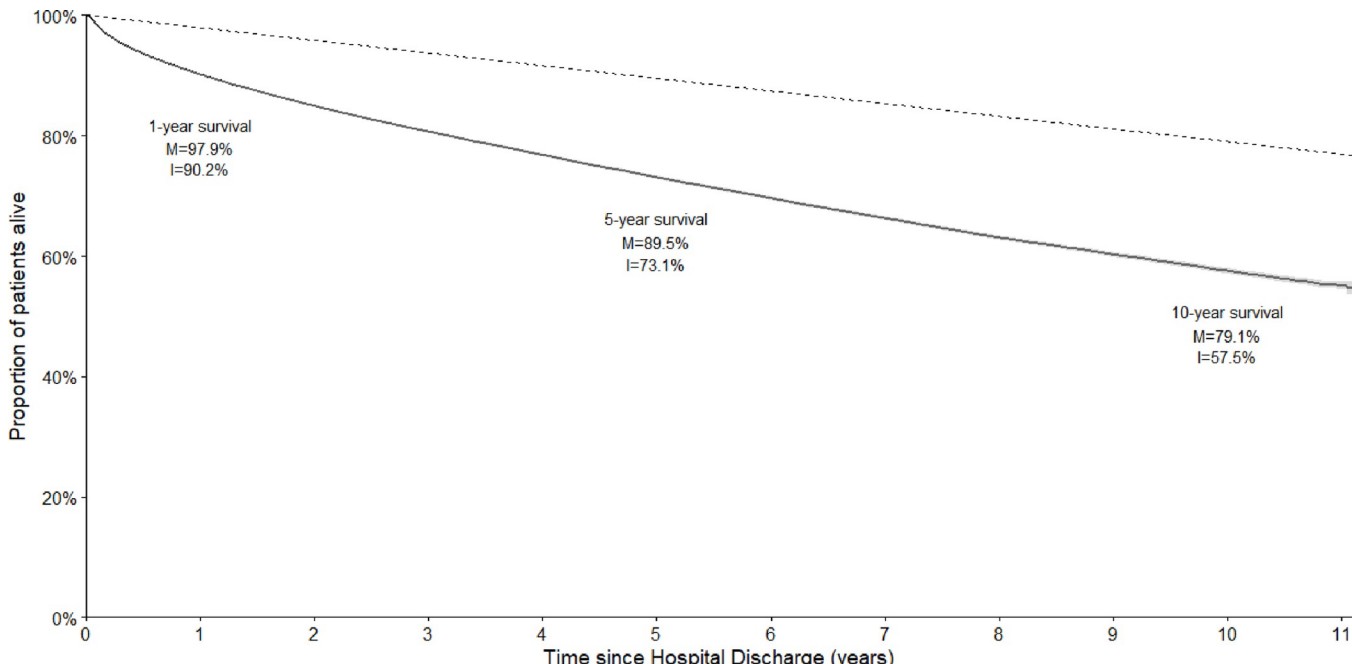

**Fig 1. Survival curve of the study cohort (solid line) compared to the matched cohort (dotted line) (N = 130707).** I; Intensive Care cohort, M; Matched cohort. The 95% confidence interval of the study cohort line is represented by the shaded area.

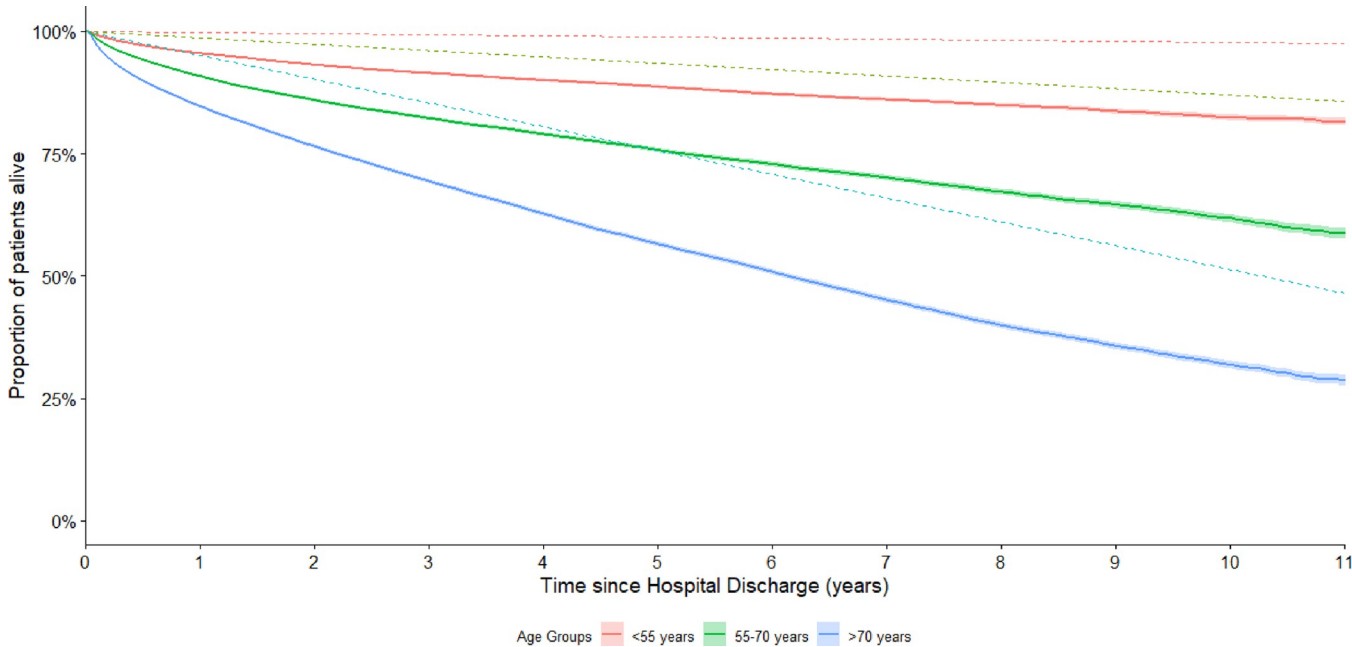

**Fig 2. Survival curve of the study cohort stratified by age groups (solid lines) compared to the matched cohorts (dotted lines) (N = 130707).** The 95% confidence intervals of the study cohort lines are represented by the shaded areas.

### Factors associated with long-term survival

A total of 111,989 patients were included in the multivariable Cox regression, as patients with missing data in any variable were excluded. Being female, younger, being partnered, having a lower comorbidity score, and coming from an area of a higher socioeconomic status were all associated with better long-term survival (*Table 2*). A shorter LOS in hospital, being invasively ventilated during the admission, and being discharged home were also associated with better long-term survival. Being discharged following an ICU admission in more recent years was associated with better long-term survival, however, it is unclear if these improvements differ from the improvements in survival see in the general population (*S2 Fig*). Compared to those admitted for cardiac surgery, all diagnoses had worse long-term survival except for non-head trauma and sub-arachnoid haemorrhages in whom there was no difference, and head trauma in which survival was better. It should be noted the hazard ratio for head trauma was only 0.87 and p-value 0.049. Those with COPD had the worst long-term survival followed by the Vascular / Thoracic Surgery group.

## Discussion

### Overview

Long-term survival of patients that initially survive admission to an Australian ICU is worse than the general age-matched population but has improved significantly over the study period. All diagnostic groups had worse long-term survival, except for elderly cardiac surgery patients who had better or equal survival. In addition, multiple demographic, socio-economic, diagnostic, acute and chronic illness factors appeared to influence long-term survival.

### Comparison with existing literature

At 1-year post-discharge 90.4% of the cohort were alive, whereas at the same time survival was 97.8% for the age-matched cohort. This figure is similar to those described in previous studies

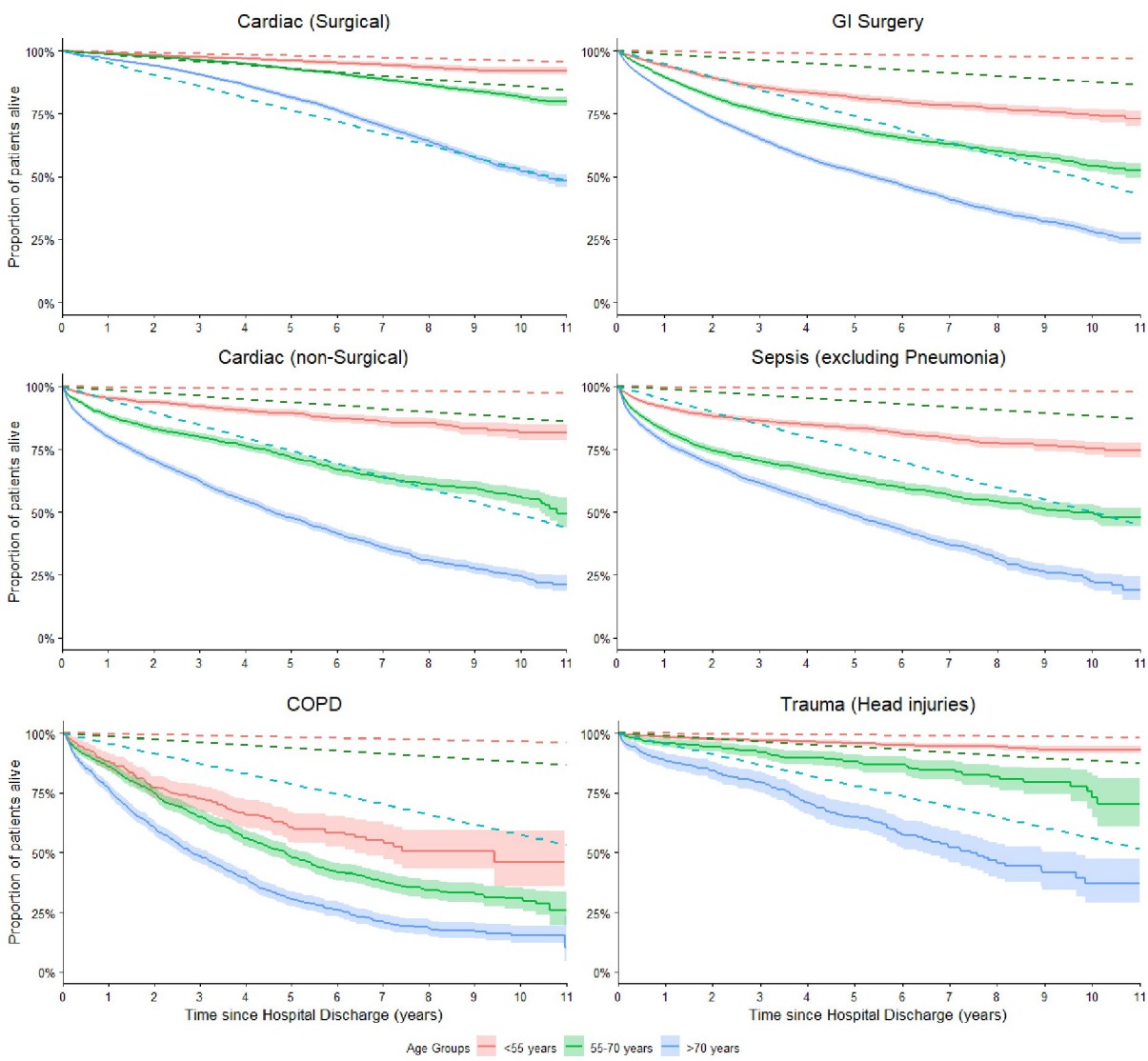

**Fig 3. Survival curves of diagnosis groups stratified by age group (solid lines) compared to the matched standard Australian population (dotted lines).** The 95% confidence intervals of the study cohort lines are represented by the shaded areas. Groups: Cardiac (Surgical) (N = 14438), GI Surgery (N = 14450), Cardiac (non-Surgical (N = 8271), Sepsis (excl. Pneumonia) (N = 8581), COPD (N = 2608), Trauma (Head injuries) (N = 2725). GI; Gastrointestinal, COPD; Chronic Obstructive Pulmonary Disease.

of ICU patients [5, 8, 20]. These studies also found the survival of age-matched general population cohorts to be higher than the ICU cohorts. When we examined up to 11-years post-discharge this difference in survival persisted across all age groups. Our finding that the older age group in the cardiac surgery group had better survival than their age-matched cohort has not been previously described and may result from heterogeneity in that candidates considered particularly robust may be more likely to undergo cardiac surgery. Long-term survival for this group has only been shown to be equivalent or worse than their age-matched cohort [21–23].

Our findings that increased long-term survival was associated with younger age, female sex, and being from a higher SES area were consistent with previous literature [3, 24]. A longer hospital LOS and being discharged to a location other than home being associated with worse long-term survival were also consistent with previous literature [3, 25, 26]. Previous work by Iwashyna et al. found that for those with a long ICU LOS (>10 days), pre-ICU patient

**Table 2. Multivariate Cox regression for long-term mortality hazard.** (N = 111989).

| Characteristic | HR | 95% CI | p-value |
|---|---|---|---|
| **Age** | | | |
| <55 | - | - | - |
| 55–70 | 2.10 | 2.03, 2.19 | <0.001 |
| >70 | 3.69 | 3.56, 3.83 | <0.001 |
| **Sex** | | | |
| Male | - | - | - |
| Female | 0.81 | 0.79, 0.83 | <0.001 |
| **Relationship status** | | | |
| Partnered | - | - | - |
| Not partnered | 1.14 | 1.12, 1.17 | <0.001 |
| **IRSAD score quartile** | | | |
| Lowest 25% | - | - | - |
| 2nd 25% | 0.98 | 0.95, 1.01 | 0.2 |
| 3rd 25% | 0.95 | 0.92, 0.98 | 0.004 |
| Highest 25% | 0.92 | 0.89, 0.95 | <0.001 |
| **Residence** | | | |
| Major City | - | - | - |
| Inner Regional | 1.03 | 1.00, 1.06 | 0.047 |
| Outer Regional | 1.09 | 1.04, 1.14 | <0.001 |
| Remote | 0.95 | 0.67, 1.33 | 0.8 |
| **Elixhauser score (+1 point)** | 1.04 | 1.04, 1.04 | <0.001 |
| **Admission post Medical Emergency Team call** | | | |
| No | - | - | - |
| Yes | 1.01 | 0.97, 1.06 | 0.5 |
| **Limitations of medical therapy at ICU admission** | | | |
| No limitations | - | - | - |
| Limitations | 1.94 | 1.86, 2.03 | <0.001 |
| Unknown | 1.08 | 0.89, 1.31 | 0.4 |
| **Year of admission** | | | |
| 2007–2009 | - | - | - |
| 2010–2012 | 0.94 | 0.91, 0.97 | <0.001 |
| 2013–2015 | 0.91 | 0.88, 0.94 | <0.001 |
| 2016–2018 | 0.87 | 0.83, 0.91 | <0.001 |
| **Hospital type** | | | |
| Tertiary | - | - | - |
| Metropolitan | 0.96 | 0.93, 0.98 | 0.002 |
| Rural / Regional | 0.91 | 0.87, 0.95 | <0.001 |
| **Admission type** | | | |
| Emergency | - | - | - |
| Elective | 1.03 | 0.99, 1.06 | 0.15 |
| **Admitted to ICU from** | | | |
| Theatre | - | - | - |
| ED | 1.00 | 0.95, 1.05 | 0.9 |
| Ward | 1.11 | 1.05, 1.18 | <0.001 |
| Hospital Transfer | 0.93 | 0.87, 0.98 | 0.011 |
| **Hospital LOS** | | | |
| <14 days | - | - | - |

(*Continued*)

**Table 2.** (Continued)

| Characteristic | HR | 95% CI | p-value |
|---|---|---|---|
| 14 days or more | 1.20 | 1.17, 1.23 | <0.001 |
| **Invasively ventilated during admission** | | | |
| No | - | - | - |
| Yes | 0.81 | 0.79, 0.84 | <0.001 |
| **CRRT during admission** | | | |
| No | - | - | - |
| Yes | 0.92 | 0.87, 0.98 | 0.006 |
| **APACHE-IIIJ physiological sub-score quartile** | | | |
| 1st 25% | - | - | - |
| 2nd 25% | 1.22 | 1.18, 1.27 | <0.001 |
| 3rd 25% | 1.41 | 1.36, 1.46 | <0.001 |
| 4th 25% | 1.55 | 1.49, 1.61 | <0.001 |
| **Discharged home** | | | |
| Yes | - | - | - |
| No | 1.45 | 1.41, 1.49 | <0.001 |
| **Diagnosis on admission to ICU** | | | |
| Cardiac (Surgical) | - | - | - |
| Cardiac (non-Surgical) | 2.02 | 1.87, 2.18 | <0.001 |
| Cardiac arrest | 1.65 | 1.48, 1.82 | <0.001 |
| Vascular/Thoracic surgery | 2.45 | 2.31, 2.60 | <0.001 |
| Abdominal aortic aneurysm | 1.55 | 1.42, 1.70 | <0.001 |
| Respiratory | 2.08 | 1.91, 2.26 | <0.001 |
| COPD | 3.03 | 2.78, 3.31 | <0.001 |
| Pneumonia | 2.01 | 1.85, 2.18 | <0.001 |
| Sepsis (excluding Pneumonia) | 1.99 | 1.85, 2.15 | <0.001 |
| Neurological | 2.20 | 2.02, 2.39 | <0.001 |
| Stroke / Intra-cerebral Haemorrhage | 1.77 | 1.57, 2.00 | <0.001 |
| Sub-arachnoid Haemorrhage | 1.03 | 0.86, 1.24 | 0.7 |
| Seizure | 1.90 | 1.69, 2.14 | <0.001 |
| Orthopaedic (non-spinal) | 2.03 | 1.88, 2.19 | <0.001 |
| Orthopaedic (spinal) | 1.87 | 1.59, 2.20 | <0.001 |
| Trauma (head) | 0.87 | 0.75, 1.00 | 0.049 |
| Trauma (non-head) | 1.10 | 0.99, 1.22 | 0.073 |
| Toxicological (overdose) | 1.88 | 1.69, 2.09 | <0.001 |
| GI Surgery | 2.00 | 1.89, 2.12 | <0.001 |
| GI Medical | 2.22 | 2.04, 2.42 | <0.001 |
| Surgical (other) | 1.81 | 1.69, 1.93 | <0.001 |
| Medical (other) | 1.89 | 1.74, 2.05 | <0.001 |

Patients with any missing data for any included variables were excluded.

HR; Hazard Ratio, CI; Confidence Interval, Ref; Reference Group, IRSAD; Index of Relative Socioeconomic Advantage and Disadvantage, ICU; Intensive Care Unit, ED; Emergency Department, LOS; Length of stay, CRRT; Continuous Renal Replacement Therapy, APACHE; Acute Physiological and Chronic Health Evaluation, COPD; Chronic Obstructive Pulmonary Disease, GI; Gastrointestinal.

characteristics were more predictive of survival to discharge than the characteristics of the admission itself [27]. Our study found that both pre-ICU and admission characteristics impacted long-term survival.

Relative to those admitted for cardiac surgery, all diagnoses except traumatic injuries and sub-arachnoid haemorrhages were associated with worse long-term survival, after adjusting for confounders. Those admitted for trauma have been previously shown to have good long-term survival, given they survive to hospital discharge [28, 29]. The finding that those with sub-arachnoid haemorrhage had similar survival to the reference group of cardiac surgery, was unexpected and should be an area of future study. Similarly, the association of invasive ventilation and renal replacement therapy with good long-term survival was also unexpected and may be an example of survivorship bias as our study only included patients that left hospital alive.

## Implications of findings

Admission to ICU is associated with a persistent and significant impact on a person's long-term survival. This study provides clinicians with objective information to inform discussions with patients and family members about the implications of admission to ICU and long-term survival after critical illness. Outcomes vary widely between patient groups, particularly when considering the reason for admission. This highlights the need to provide objective information that is patient specific during these above-mentioned discussions. The finding that those from areas of lower socioeconomic status have worse long-term survival demonstrates the need to carefully consider such factors during discharge planning. Improvements in the long-term survival of patients over time may indicate changes in patient selection or improvement in therapeutic outcomes with lasting benefit This observation requires further validation.

## Strengths

Our study had multiple strengths. Firstly, using an established database, the ANZICS APD, we were able to include admissions to all 23 public ICUs in Victoria. These units admit patients from an area that is home to approximately 25% of the Australian population [16]. This limited the selection bias that impacts many studies that only include only small numbers of hospitals. Secondly, our follow-up period was 11.1 years, with a median of 3.6 years. This is significantly longer than many other comparable studies, providing a true long-term description of survival [4, 20]. Finally, through using multiple databases we were able to look at not only factors relating to that patient's admission, but also personal factors such as relationship status, the patient's area of residence and socioeconomic status. Such factors have been previously identified as 'research gaps' [8].

## Limitations

Several limitations must be considered. Firstly, the linkage between the ANZICS APD and the VAED was only 80.3%. However, there were few clinically significant differences between the linked and unlinked groups, and both groups had identical median APACHE III-J scores. Probabilistic linkage was employed as the ANZICS APD did not contain patient identifying information such as name and date of birth. Secondly, the Victorian Death Registry only records deaths that occurred in the study state, this results in any patients that died outside of Victoria being recorded as alive. To mitigate against this all patients that had an area of residence outside of Victoria were excluded. Thirdly, the study only included admissions to ICUs in public hospitals. Patients admitted to ICUs in private hospitals are more commonly elective surgical cases, with a lower rate of invasive ventilation, therefore care should be taken when extrapolating our results to this population [17]. Fourthly, data on socioeconomic status was the average for the area the patient resided in and was not specific to the individual patient. Fifthly, given that those admitted to ICU in more recent years had better survival, if this trend

were to continue our data would have underestimated the long-term survival of patients currently being admitted to ICU. Sixthly, our study only reported survival. Previous work has found high rates of self-reported disability and low rates of returning to work at six-months post ICU admission [30, 31]. Therefore, it is important to consider that survival does not necessarily equate with an acceptable quality of life for all patients. Seventhly, we did not capture data regarding ongoing care provided to patients beyond discharge, except for whether patients were discharged to sites other than home. Finally, it must be considered by all readers that our cohort may be different from the patients they care for, and the whole group outcomes may differ significantly to the outcomes for specific patient cohorts. This is particularly relevant for countries with different healthcare systems to Australia. Of note our rates of mechanical ventilation and renal replacement therapy were low. The fact our study only included patients that survived to hospital discharge is likely a contributor to this.

### Future directions

Future work should focus on describing long-term outcomes with measures that quantify quality of life and disability in addition to mortality, and also assess the impact of discharge to locations other than home such as rehabilitation and long-term care facilities. Further analysis of the cardiac surgery diagnosis group is needed. Work should focus particularly on the elderly in this group to provide insight into why they had better long-term survival than both the other diagnosis groups and their age-matched cohort. Additionally, investigation into whether the superior survival outcomes of this group extend to quality of life is warranted.

### Conclusions

Patients admitted to ICU in Victoria, Australia who survive to discharge have lower long-term survival than that of the general population across all age groups for at least the first 11 years post-discharge. At 1-year post discharge 90.2% of patients were alive, with this decreasing to 57.7% after 10-years post discharge. This poor comparative survival did not extend to elderly patients undergoing cardiac surgery, who had better or equal survival than their matched general population cohort. These data should be considered when discussing the implications of an ICU admission with patients and their families, including goals of care.

### Supporting information

**S1 Fig. Inclusion and exclusion process.** * This criterion requires a variable only available for matched patients. Its calculation was performed prior to the researchers receiving receipt of the data. Therefore, if an admission for an individual was one of the excluded unmatched admissions it would still have been considered when this variable was created. ANZICS; Australian and New Zealand Intensive Care Society, VAED; Victorian Admitted Episode Dataset.
(DOCX)

**S2 Fig. Survival curves limited to 3-years post discharge stratified by year group (solid lines) compared to the matched standard Australian population (dotted lines).** Follow-up is only 3-years post hospital discharge to allow comparison between each year period. Three-year period groups were created to facilitate simple comparisons.
(DOCX)

**S3 Fig. Survival curves of stratified by readmission status or admission type (solid lines) compared to the matched standard Australian population (dotted lines).** The 95% confidence intervals of the study cohort lines are represented by the shaded areas. Groups:

Readmissions (N = 130707), Admission type (N = 129830).
(DOCX)

**S1 Table. Comparison of characteristics between patients linked to the VAED and those not able to be linked.** The unlinked cohort had the same inclusion criteria applied to it as the linked cohort. Variables that appeared in the *Table 1* of the main manuscript that are absent from this table were those not included in both datasets. IQR; Interquartile range, Index of Relative Socioeconomic Advantage and Disadvantage, ICU; Intensive Care Unit, ED; Emergency Department, LOS; Length of stay, APACHE; Acute Physiological and Chronic Health Evaluation, COPD; Chronic Obstructive Pulmonary Disease, GI; Gastrointestinal.
(DOCX)

## Acknowledgments

The authors would like to acknowledge all persons and units involved in the submission of data to the ANZICS APD, the VAED and the Victorian Death Registry.

## Author Contributions

**Conceptualization:** Zakary Doherty, David Pilcher.

**Data curation:** Zakary Doherty, David Bevan, Sharon Williams, Andrew Wilson, David Pilcher.

**Formal analysis:** Zakary Doherty, Graeme Duke, David Pilcher.

**Investigation:** Rebecca Kippen.

**Visualization:** Zakary Doherty.

**Writing – original draft:** Zakary Doherty.

**Writing – review & editing:** Zakary Doherty, Rebecca Kippen, David Bevan, Graeme Duke, Sharon Williams, Andrew Wilson, David Pilcher.

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
