## [Decision Letter · Decision Letter 0]

19 Oct 2021

PONE-D-21-27970Long-term outcomes of Intensive Care survivors in Australia: a multi-centre retrospective cohort studyPLOS ONE

Dear Dr. Doherty,

Thank you for submitting your manuscript to PLOS ONE. After careful consideration, we feel that it has merit but does not fully meet PLOS ONE’s publication criteria as it currently stands. Therefore, we invite you to submit a revised version of the manuscript that addresses the points raised during the review process.

ACADEMIC EDITOR: 

The manuscript is interesting but will require further reworking and a major revision.<o:p></o:p>

While they recognize the potential interest of the subject studied, the reviewers raised a several of important issues that need to be properly addressed.

We look forward to receiving your revised manuscript.

Kind regards,

Marcelo Arruda Nakazone, M.D., Ph.D.

Academic Editor

PLOS ONE

Journal Requirements:

Reviewers' comments:

Reviewer's Responses to Questions

**Comments to the Author**

1. Is the manuscript technically sound, and do the data support the conclusions?

Reviewer #1: Yes

Reviewer #2: Yes

Reviewer #3: Yes

Reviewer #4: Partly

2. Has the statistical analysis been performed appropriately and rigorously? 

Reviewer #1: Yes

Reviewer #2: Yes

Reviewer #3: Yes

Reviewer #4: Yes

3. Have the authors made all data underlying the findings in their manuscript fully available?

Reviewer #1: Yes

Reviewer #2: Yes

Reviewer #3: No

Reviewer #4: Yes

4. Is the manuscript presented in an intelligible fashion and written in standard English?

Reviewer #1: Yes

Reviewer #2: Yes

Reviewer #3: Yes

Reviewer #4: Yes

5. Review Comments to the Author

Reviewer #1: Thank you for giving me the opportunity to review the paper “Long-term outcomes of Intensive Care survivors in Australia: a multi-centre retrospective cohort study” by Doherty et al.

Comments: This is an interesting study to show the long-term outcomes of intensive care survivors. However, there are some concerns about this study.

1) The length of stay in ICU seems to be short, can you add the data of SOFA score as organ damage at the time of ICU admission?

2) Do the authors have any data about the length of ventilation? Please add the data in Table 1 and 2.

3) Most of the patients discharged from ICU have lower quality of life due to ICU-acquired weakness and are often transferred to rehabilitation facilities or chronic care hospitals. What do you think is the reason why the discharge rate to home is so high in this study?

4) I recommend that the following three suggestions to make the Table 1 and 2 easier to read: to delete the ruled line surrounding the title of Table 1 and 2, to delete the vertical line in the table, and to delete all horizontal lines of the table but the first, second, and bottom.

Reviewer #2: This study describes long-term (median follow-up 3.6 years) outcome of survivors after an ICU stay in Australia. The main result is that the survival rates for ICU survivors are consistently worse than those for general population, except for patients over 70 years with cardiac surgery. These results have been previously reported but the strength of this study is the size of the studied sample and the analysis of a number of potentially explanatory factors, usually not taken into account.

I have only minor comments or questions:

- considering that all ICU stays could be included, the ratio of elective admissions seems normal but I am surprised by the short reported median ICU length of stay and the quite low use rates of mechanical ventilation (MV) and renal replacement (CRRT). A comment about these points could be added.

- although much less used than MV, CRRT seems also related to a protective effect like MV. The authors could more comment on these seemingly paradoxical results.

- the authors performed analysis of multiple admission diagnosis sub groups but an analysis of the outcome of overall medical and overall surgical patients should be also interesting. In the same way, comparison of the outcome of patients with multiple ICU stays vs single stay should be performed.

- the study address only survival and not quality of life or disabilty. This point is stated by the authors but could be more highlighted.

- as stated by the authors, results of the study should be considered when discussing an ICU admission; however these results are derived from a large sample (which is a strength), with differences between sub groups related to some factors. So, it seems reasonable to be cautious in the face of excessive generalisation, particularly in case of hyperspecialized center or very particular type of patients.

Reviewer #3: GENERAL COMMENTS

In this retrospective study on mortality after ICU discharge, the authors compared 130,775 “ICU survivors” (see comment below) mortality with expected survival for the general population matched on year, age and sex to the study cohort. The general conclusion is that “ICU survivors” have worse long-term survival than the general population (except for the elderly admitted to ICU following cardiac surgery).

The authors should be congratulated for the amount of work and the large dataset they were able to build. The authors have a huge collection of data that are certainly the source for future manuscripts. The study is well written and already acknowledges several limitations.

I would like to raise the next comments:

SPECIFIC COMMENTS

The way I understand the study is that only patients who left the hospital were evaluated for the long-term outcome. However, there were certainly patients who left the ICU but died in hospital. I could not find this information, which I believe, is of importance. Hence, the title of the manuscript is not really adequate; a title such as “Long-term outcomes of patients who left the hospital after an ICU stay” looks more appropriate.

The authors report on just under half of patients were invasively ventilated (line 240-241). Does this include elective post-operative patients? If so, they should be removed as this MV does not really match those on MV bc of respiratoy failure or other critical condition.

This might affect what is written in the discussion section (line 359-361) and the result in table 2.

The authors compare mortality between “ICU survivors” and an aged-matched population. Not really surprising as those admitted in ICU probably suffer from more comorbidities. This is perfectly demonstrated with the COPD burden.

The reason to take cardiac surgical patients as the reference for the Multivariate Cox regression for long-term mortality hazard remains unclear.

Likewise, some of the following results are quite difficult to understand: for instance, why would head trauma patients display lower long term mortality thatn cardiac surgical patients? Are they younger?

Reviewer #4: Dear authors

first of all a very interesting work. However I have sone concerns listed below:

The statements of this very interesting results should be reconsidered to the limitations of the study setting as mentioned below.

Study population

This cohort represents 53,5% of ICU admissions of the period! This should be clarified in the manuscript. One point is, that 6,25% didn´t survive the hospital stay and 16,8% were admitted for a second ICU stay it has to be pointed out that 21% of the patients could not be matched. Therefore, the strengths of this very good and important study is limited as authors demonstrate in table S1 that in all parts values are significantly different between matched and not matched cases. Therefore generalisability should be

Line 359 “Similarly, the association of invasive ventilation and good long-term survival was also unexpected and may be an example of survivorship bias as our study only included patients that left hospital alive.” For this statement it would be useful to look on the data, how many of these ventilated patients came from OR after elective surgery, as 48% of patients where admitted from OR. These patients have a priori a better chance of survival as documented by APACHE or SAPS score demonstrate.

Line 365 “This study provides clinicians with objective information to inform discussions with patients and family members about the implications of admission to ICU and long-term survival after critical illness.” This statement has to been seen with concern. The study data provide a hint, if patients survive hospital admission, what may be the future. There several limitations of the data matched for this analysis (e.g. see point one). Therefore, it may be useful to look on the patients who not survived ICU stay, according to the underlying diseases. Further Australia has its own way, whether patients will be admitted to ICU, or not. Meaning, there are many patients admitted to ICU in other regions of the world who would never be admitted to ICU in Australia.

Line 382 “Secondly, our follow-up period was 11.1 years, with a median of 3.6 years.” This should be put into respect, that this is only correct for patients admitted in 2007 and not for the hole study population.

6. PLOS authors have the option to publish the peer review history of their article (what does this mean?). If published, this will include your full peer review and any attached files.

Reviewer #1: No

Reviewer #2: **Yes: **Emmanuel Guerot

Reviewer #3: No

Reviewer #4: No

---

## [Author Response · Author response to Decision Letter 0]

29 Dec 2021

Please see attached word document with tabulated responses to all comments.

---

## [Decision Letter · Decision Letter 1]

14 Mar 2022

Long-term outcomes of hospital survivors following an ICU stay: a multi-centre retrospective cohort study

PONE-D-21-27970R1

Dear Dr. Doherty,

We’re pleased to inform you that your manuscript has been judged scientifically suitable for publication and will be formally accepted for publication once it meets all outstanding technical requirements.

Kind regards,

Marcelo Arruda Nakazone, M.D., Ph.D.

Academic Editor

PLOS ONE

Additional Editor Comments (optional):

Reviewers' comments:

Reviewer's Responses to Questions

**Comments to the Author**

1. If the authors have adequately addressed your comments raised in a previous round of review and you feel that this manuscript is now acceptable for publication, you may indicate that here to bypass the “Comments to the Author” section, enter your conflict of interest statement in the “Confidential to Editor” section, and submit your "Accept" recommendation.

Reviewer #1: All comments have been addressed

Reviewer #2: All comments have been addressed

Reviewer #3: All comments have been addressed

2. Is the manuscript technically sound, and do the data support the conclusions?

Reviewer #1: Yes

Reviewer #2: Yes

Reviewer #3: Yes

3. Has the statistical analysis been performed appropriately and rigorously? 

Reviewer #1: Yes

Reviewer #2: Yes

Reviewer #3: Yes

4. Have the authors made all data underlying the findings in their manuscript fully available?

Reviewer #1: Yes

Reviewer #2: Yes

Reviewer #3: Yes

5. Is the manuscript presented in an intelligible fashion and written in standard English?

Reviewer #1: Yes

Reviewer #2: Yes

Reviewer #3: Yes

6. Review Comments to the Author

Reviewer #1: (No Response)

Reviewer #2: (No Response)

Reviewer #3: In this retrospective study on mortality after ICU discharge, the authors compared 130,775 “ICU survivors” (see comment below) mortality with expected survival for the general population matched on year, age and sex to the study cohort. The general conclusion is that “ICU survivors” have worse long-term survival than the general population (except for the elderly admitted to ICU following cardiac surgery).

I thank the authors for taking into account the comments I rose (as well as those from the other reviewers).

I have no further comments.

7. PLOS authors have the option to publish the peer review history of their article (what does this mean?). If published, this will include your full peer review and any attached files.

Reviewer #1: No

Reviewer #2: **Yes: **Emmanuel Guerot

Reviewer #3: No

---

## [Editor Report · Acceptance letter]

18 Mar 2022

PONE-D-21-27970R1 

Long-term outcomes of hospital survivors following an ICU stay: a multi-centre retrospective cohort study 

Dear Dr. Doherty:

I'm pleased to inform you that your manuscript has been deemed suitable for publication in PLOS ONE. Congratulations! Your manuscript is now with our production department. 

Kind regards, 

on behalf of

Professor Marcelo Arruda Nakazone 

Academic Editor

PLOS ONE